# Comparison of Post-Vaccination Cellular Immune Response in Patients with Common Variable Immune Deficiency

**DOI:** 10.3390/vaccines12080843

**Published:** 2024-07-25

**Authors:** Aristitsa Mikhailovna Kostinova, Elena Alexandrovna Latysheva, Mikhail Petrovich Kostinov, Nelly Kimovna Akhmatova, Svetlana Anatolyevna Skhodova, Anna Egorovna Vlasenko, Alexander Petrovich Cherdantsev, Irina Leonidovna Soloveva, Isabella Abramovna Khrapunova, Marina Nikolaevna Loktionova, Ekaterina Alexandrovna Khromova, Arseniy Alexandrovich Poddubikov

**Affiliations:** 1Federal State Autonomous Educational Institution, Higher Education I.M. Sechenov First Moscow State Medical University of the Ministry of Health of the Russian Federation (Sechenov University), Trubetskaya Str., 8/2, 119991 Moscow, Russia; 2National Research Center Institute of Immunology Federal Medical-Biological Agency of Russia, Kashirskoe Shosse, 24, 115478 Moscow, Russia; 3Faculty of Medicine and Biology, Pirogov Russian National Research Medical University, Ostrovitianov Str., 1, 117513 Moscow, Russia; 4Federal State Budgetary Scientific Institution «I.I. Mechnikov Research Institute of Vaccines and Sera», Malyi Kazenniy Pereulok, 5a, 105064 Moscow, Russia; 5Federal State Budgetary Educational Institution, Higher Education “Samara State Medical University” of the Ministry of Healthcare of the Russian Federation, Chapaevskaya Street, 89, 443099 Samara, Russia; 6Federal State-Funded Educational Institution, Higher Education “Ulyanovsk State University”, Leo Tolstoy Street, 42, 432017 Ulyanovsk, Russia; a_cherdantsev@rambler.ru (A.P.C.);; 7Federal Budget Institute of Science “Central Research Institute of Epidemiology” of the Federal Service for Surveillance on Consumer Rights Protection and Human Wellbeing, st. Novogireevskaya, 3a, 111123 Moscow, Russia

**Keywords:** adjuvanted influenza vaccines, CVID, toll-like receptors, influenza, azoximer bromide

## Abstract

Background: The problem of identifying vaccine-specific T-cell responses is still a matter of debate. Currently, there are no universal, clearly defined, agreed upon criteria for assessing the effectiveness of vaccinations and their immunogenicity for the cellular component of immunity, even for healthy people. But for patients with inborn errors of immunity (IEI), especially those with antibody deficiencies, evaluating cellular immunity holds significant importance. Aim: To examine the effect of one and two doses of inactivated adjuvanted subunit influenza vaccines on the expression of endosomal Toll-like receptors (TLRs) on the immune cells and the primary lymphocyte subpopulations in patients with common variable immunodeficiency (CVID). Materials and methods: During 2018–2019, six CVID patients received one dose of a quadrivalent adjuvanted influenza vaccine; in 2019–2020, nine patients were vaccinated with two doses of a trivalent inactivated influenza vaccine. The proportion of key lymphocyte subpopulations and expression levels of TLRs were analyzed using flow cytometry with monoclonal antibodies. Results: No statistically significant alterations in the absolute values of the main lymphocyte subpopulations were observed in CVID patients before or after vaccination with the different immunization protocols. However, after vaccination, a higher expression of TLR3 and TLR9 in granulocytes, monocytes, and lymphocytes was found in those patients who received two vaccine doses rather than one single dose. Conclusion: This study marks the first instance of using a simultaneous two-dose vaccination, which is associated with an elevated level of TLR expression in the immune cells. Administration of the adjuvanted vaccines in CVID patients appears promising. Further research into their impact on innate immunity and the development of more effective vaccination regimens is warranted.

## 1. Introduction

### 1.1. Actuality

Primary immunodeficiencies (PIDs) or inborn errors of immunity (IEI) are a group of orphan diseases that are clinically mainly represented by recurrent, severe, or atypical infections [1,2]. The main cohort of patients over 18 years old with IEI are patients with impaired antibody synthesis; up to 20% of patients from this group—with a defect in the humoral immune system—are individuals with common variable immune deficiency (CVID) [3]. The main cause of hospitalization, disability, and premature death of these patients is postinfectious complications due to an impaired ability to synthesize antibodies in response to infection as well as to vaccination. That is why until the beginning of the 21st century, it was believed that vaccination of this cohort of patients was unsafe and inappropriate; furthermore, for some forms of PID, it is even a diagnostic criterion [4]. This position changed drastically in 2014, when the first position paper was published in which vaccination with inactivated drugs was recommended for all patients with primary immunodeficiency diseases, including those with impaired antibody synthesis [5,6]. However, the evidence base is insufficiently presented due to the rarity of occurrence of this pathology in the population as well as the unequal ability of patients to synthesize antibodies due to the heterogenicity of defects inside the group itself [3,7]. At the same time, due to the growing awareness of medical practitioners, improvements in routine diagnostic methods, and increased availability of laboratory testing, there is a rapid increase in the detection of patients with this diagnosis [8]. Therefore, the need and interest in studying the effectiveness of vaccination is topical and real for the whole medical community (as they can be under the supervision of specialized doctors for a long time), with the goal of improving the quality of life of such patients.

Due to the rarity of this pathology, all currently available studies on the effectiveness of influenza virus vaccination (IVV) as an annual “threat” to the population during the cold period of the year among adult patients with CVID were conducted on a very small group, often heterogeneous, with an average number of up to 10 people [9,10,11,12,13,14]. Another difficulty in studying the effectiveness of vaccination is the annual change in circulating strains in different hemispheres, and consequently different strain composition of seasonal influenza virus vaccines. The components of the vaccines also differ.

Moreover, the markers for evaluation of cellular post-vaccination immunity (humoral immune response is significantly weakened in CVID patients) are not still defined even for healthy people. The first and still only attempts to identify criteria were made in 2018 by the European Medicines Agency, when in addition to assessing the immunogenicity of vaccines by geometric mean antibody titers (GMTs) and their pre-to-post-vaccination ratio (GMR), seroconversion rates and seroprotection levels, two more criteria for the assessment of cellular and cell-mediated immunity, were first identified, which are carried out to a lesser extent in routine practice, and some are not available at all:assessment of the cell-mediated component of the immune response (for example, by quantification of T lymphocytes specific for vaccine antigen(s) and/or antigens isolated from wild-type microorganisms in vitro by direct marker incorporation or based on cytokine release);assessment of cellular immunity: number and proportion of participants before and after vaccination with sensitized (i.e., antigen specific) T lymphocytes (including sensitized CD4+ and CD8+ T lymphocytes) depending on the antigenic substance(s) used for stimulation and cytokines detected during the assay(s).

Since 2009, in order to increase the immunogenicity of vaccines to stimulate the synthesis of antibodies in the early stages after administration of the drug, adjuvanted vaccines have begun being used, which have an activating effect on the cellular component of immunity that is especially important for enhancing the immune response in immunocompromised patients [15,16,17,18,19,20]. In this regard, even studies with a small number of participants with CVID play an important role in generating knowledge for the medical community as a whole and contribute to the study and exchange of experience in matters of patient vaccination through the expansion and creation of international databases and registries.

### 1.2. Cellular Immunity in Patients with CVID

In the structure of IEI, immunodeficiency with impaired humoral immunity represents about 50% of all nosologies. Common variable immune deficiency (CVID) is the most common variant of primary immunodeficiency, affecting predominantly the humoral immune system. CVID is characterized by various clinical manifestations, among which the most common are recurrent infections caused by hypogammaglobulinemia, impaired production of specific antibodies in response to protein and polysaccharide antigens, and manifestations of immune dysregulation [21]. Symptoms of immune dysregulation include autoimmune complications, enteropathy, pathological lymphoproliferation, and a several hundred-fold increased risk of malignancy, primarily lymphoma [22,23,24].

The etiology of CVID is still unestablished. Genetic alterations are identifiable as the origin of disease in around 10–20% of patients, contingent on ethnicity and study design [25]. Novel monogenic defects exhibit traits similar to CVID [26,27]. It is possible that the immune mechanism in response to vaccination and infection depends on the type of monogenic defect. Nonetheless, molecular genetic testing has been conducted in fewer than 20% of CVID patients.

Heated debate about IVV of patients undergoing regular re-placement therapy with intravenous or subcutaneous immunoglobulin preparations arises annually. In patients with CVID with an impaired antibody formation, despite regular replacement therapy with donor immunoglobulins, being vaccinated against influenza is essential for anti-virus protection avoiding complications. This is because intravenous immunoglobulins (IVIGs) do not contain antibodies to the current strains of the influenza virus. Moreover, influenza virus vaccines can trigger cellular immunity, producing influenza-specific CD4 and CD8 T-lymphocytes in patients who cannot generate IgG antibodies against the virus, making vaccination warranted [19,28,29]. Although there is some evidence suggesting the presence of cross-reactive A/H1N1 antibodies in IVIGs [30], other studies have not confirmed this. In a study by Gardulf M. M. et al., none of the 48 CVID patients undergoing regular immunoglobulin therapy (weekly) had detectable antibodies to the influenza virus before vaccination [13].

The question of vaccination schedules for immunocompromised patients also remains open. Although Eibl M. M. and Wolf H. M. showed that IVV should be hold on once a year, the same as for non-immunocompromised individuals, other research findings indicate that a second dose should be administered no less than 21 days after the initial vaccination, or alternatively, two doses can be given at once to more effectively stimulate the immune response [5,10,11]. However, in a study by Hartley G. E. et al., it was shown that in five patients with impaired antibody synthesis, the number of antigen-specific memory B cells to hemagglutinin of the strain A/H1N1/Michigan/2015, as well as the level of predominantly IgG1, was significantly lower compared with healthy controls and did not increase after repeated vaccination [12].

Currently, there are only a few studies in the world that have estimated the formation of post-vaccination immunity, both humoral and cellular, in response to IVV in a limited number of patients with CVID [15,16,31,32,33,34].

Although an impaired adaptive immune response and defects in B-cell maturation and activation are common in the majority of patients with CVID [35], scientific attention is focused on the defects in innate immunity as a possible explanation for the heterogenicity of the group of patients with CVID. The limited available research suggests an additional role for T cells in the pathogenesis of CVID. 

Towards the end of the twentieth century, in addition to identifying B-cell abnormalities in individuals with CVID, researchers also documented various T lymphocyte dysfunctions. These included reduced lymphocyte proliferation in response to antigens, fewer naive CD4 and CD8 lymphocytes, CD4+ T-lymphopenia, an elevated count of CD8+ cells leading to an inverted CD4/CD8 ratio, a tendency toward accelerated T-cell apoptosis, diminished numbers and functions of regulatory T-lymphocytes (Tregs), increased levels of T-cell activation markers, and impaired cytokine production [32,33,34,36,37,38,39,40,41].

Some T-cell abnormalities in CVID, which include oligoclonal expansion of CD8+ T-cells and a decreased number of CD4+ T cells [42], lead to the impaired secretion of a number of soluble mediators [43].

It is believed that T-cells in CVID may be functionally exhausted and manifest as a decrease in the ability to respond to bacterial antigens [43,44]. Early studies reported signs of functional exhaustion and dysfunction of T lymphocytes [43,45,46], but to date it has been proven that their ability to produce pro-inflammatory cytokines, function, and proliferate is preserved [47]. It also follows from the results of other researchers that CD4 lymphocytes, in response to stimulation with the vaccine antigen, proliferate at a sufficient level; however, these stimuli induce significantly lower production of IL-2 than is observed among CD4+ T-cells in healthy people [48]. Serum cytokines in CVID are often shifted towards the Th1 phenotype [45,49].

The role of cytotoxic CD8+ T-lymphocytes in protection against viruses or auto-immune diseases has been well described but has not yet been studied in CVID. In patients with CVID, the percentage of activated CD8+ T-lymphocyte subsets is higher than in healthy controls. Whether this is a consequence or part of the abnormalities of CVID is still a matter of debate. Activation of CD8+ T-cells cannot always be explained by the course of a known infectious disease that can stimulate the immune system.

An altered distribution of dendritic cells (DCs) in the peripheral blood, as well as an impaired ability of DCs to activate T lymphocytes after antigenic or allogeneic stimulation in patients with CVID, has been previously described [50,51]. This is associated with a decreased expression of major histocompatibility complex class II and costimulatory molecules, as well as interleukin-12 (IL-12) synthesis by dendritic cells (DCs). Other sources have reported decreased production of IL-12 by monocytes in patients with CVID [52]. Some data have also demonstrated a decrease in circulating natural killer (NK) cell levels [53]. However, in CVID, plasmacytoid DCs (pDCs) and B cells demonstrate an impaired response to CpG stimulation in vitro [54].

### 1.3. Close Interaction of Innate and Adaptive Immunity

What is the link between B-cells and receptors of innate immunity—Toll-like receptors? Why are we talking about them in the context of cellular immunity? There is a hypothesis that TLR signaling pathways might serve as additional stimuli for the development of B cells, even though any defective signaling through these innate receptors could be compensated for by other molecular mechanisms [55]. Studies, both in vivo and in vitro, indicate that switching B cells to IgG isotypes necessitates at least two signals in addition to BCR activation: TLR activation coupled with either CD40 or IFN-alpha [56]. These findings propose that TLR activation might provide a sustained stimulus crucial for the growth and differentiation of memory B cells into mature antibody-secreting cells that are initiated by BCRs and T cells [57,58].

B cells, while primarily known for their essential role in adaptive immunity via antibody production, experience enhanced functionality and survival through the costimulatory effect of activating innate immune receptors [59]. This activation bridges innate and adaptive immune signals, prompting various cellular responses. Memory B cells are generated in germinal centers in response to either T-dependent or T-independent antigens. Similar to other antigen-presenting cells, B cells express a range of TLRs [60,61], which are persistent membrane proteins providing alternate activation routes for B cells [62].

Among these, endosomal TLRs are particularly potent in inducing B-cell activation and maturation: TLR7 interacts with single-stranded RNAs or synthetic agonists, while TLR9 responds to unmethylated CpG motifs in microbial DNA [63]. The engagement of TLR9 with CpG DNA has been shown to activate normal B cells, elevate the expression of costimulatory molecules, trigger the secretion of IL-6 and IL-10, and promote T-independent isotype switching and antibody production independent of the BCR [64,65,66,67,68]. BCR binding leads to the swift activation of their expression. The activation and interaction of TLR7 and TLR9 can initiate B-cell differentiation following antigen stimulation via the BCR. However, research indicates impairments in TLR7 and TLR9 in B cells from CVID patients [63]. Given the critical role of TLR activation in memory B-cell activation and survival [69], studies have shown that in CVID patients, these B cells are not activated by CpG ODN ligands, even when costimulated with BCR, and do not secrete IL-6 and IL-10. Consequently, there is no TLR activation, low B-cell proliferation, and a lack of maturation, isotype switching, and production of IgG and IgA [63]. Naïve B cells exhibit low levels of TLRs, whereas memory B cells initially express TLR7, TLR8, and TLR9 at higher levels [70,71,72,73]. These differences align with their distinct adaptive functions: memory B cells exhibit greater TLR expression and an enhanced ability to differentiate into plasma cells upon TLR stimulation compared to naïve B cells [74]. However, prolonged activation of pathogen-associated molecular patterns (PAMPs) is believed to potentially lead to adverse effects on the body due to the regulation of the TLR signaling pathway by various feedback mechanisms [20].

### 1.4. Adjuvanted Influenza Virus Vaccines

In recent decades, to improve the effectiveness of vaccinations, new technologies us-ing various adjuvants have been developed. One of the main advantages of adjuvanted vaccines is the ability to reduce the antigenic load in the vaccine compound without losing its immunogenic properties, which improves tolerability and significantly minimizes the risks of adverse post-vaccination events. The inclusion of adjuvants made it possible to reduce the dose of antigens, with the subsequent achievement of a protective level of specific IgG after vaccination, which are synthesized in a shorter period of time at the same or even higher level than after the administration of unadjuvanted vaccines. However, the use of adjuvants in quadrivalent vaccines is currently limited.

In accordance with the recommendations of the Joint Committee on Vaccination and Immunization, the use of trivalent adjuvanted vaccines against influenza viruses in the world is recommended for people over 65 years of age, which have advantages over non-adjuvanted quadrivalent and trivalent vaccines [73].

The trivalent influenza vaccine with the addition of adjuvant MF59C.1^®^ (9.75 mg squalene) which is used in Europe, according to the results of a meta-analysis, showed statistically significant superiority in effectiveness regardless of the influenza virus strain compared to unadjuvanted ones and high immunogenicity against heterologous strains, especially against A/H3N2 [20].

In Russia, the first adjuvanted influenza virus vaccine, Grippol, was introduced into healthcare practice in 1997. Later, in 2008, Grippol Plus was registered, featuring an amount of influenza virus reduced threefold, to 5 μg, of two influenza viruses, type A (A/H1N1 and A/H3N2) and type B antigens, along with 500 μg of azoximer bromide, without loss of its immunogenic properties. The addition of an adjuvant to the composition of the vaccine made it possible to achieve a protective immune response with less amounts of hemagglutinin compared to standard doses, and also provided more stable and long-lasting immunity due to the slower release of the drug in the body and activation of cellular mechanisms in the formation of a post-vaccination response.

Many research works have demonstrated that the immunogenicity and protective qualities of antigens linked to azoximer bromide—synthetic high-molecular-weight polymer carriers—are amplified tenfold. This combination bolsters both antibody and cell-mediated immune responses and increases the production of all immunoglobulin classes (IgM, IgG, IgA) except for IgE [75]. Studies using Grippol Plus have shown that it accelerates the maturation of dendritic cells and enhances their migration rate to regional lymph nodes. It is important because a high level of DCs is one of the factors reducing susceptibility to infectious diseases. Activation of additional receptors that are responsible for recognizing bacterial antigens was also noted, and the level of antibodies remained at a protective level longer than after nonadjuvanted vaccines, which also indicates the activation of nonspecific protective factors, including against bacterial agents [18]. Thus, the combination of influenza virus vaccine strains with an adjuvant was found to be a potent activator of B- and T-lymphocytes, a discovery that led to modifications in vaccine production and subsequent clinical use of the adjuvanted vaccines in a heterogeneous group of patients with weakened immune responses.

Between 2009 and 2019, numerous post-registration studies were conducted to evaluate the safety, immunogenicity, and efficacy of a trivalent subunit adjuvanted influenza virus vaccine in various high-risk groups. These groups included pregnant women and children, individuals aged 60 and above, people with cardiovascular conditions, and patients with chronic pulmonary diseases. The findings indicated that the vaccine demonstrated strong immunogenicity and was well tolerated across all the cohorts, in alignment with the Russian national immunization program [76,77]. Additionally, clinical investigations confirmed that the adjuvanted vaccine was tolerated without adverse effects on fetal or child development [76,77,78,79,80].

## 2. Materials and Methods

### 2.1. Aim

This paper aimed to study the effects after one- and two-dose administration of inactivated adjuvanted subunit influenza virus vaccines on the expression of endosomal Toll-like receptors on immunocompetent cells and the main lymphocyte subpopulations in patients with CVID.

### 2.2. Participants

Conducted in 2018, this research analyzed 297 outpatient records of individuals with IEI, particularly CVID, from the Institute of Immunology registry, which represents a nationwide patient cohort in Russia. The diagnosis of CVID adhered to the criteria set by the European Society for Immunodeficiency Diseases (ESIDs). However, due to strict inclusion criteria and the need to manage exacerbations in hospitalized patients, coupled with annual variations in influenza virus vaccine strains and the timing of vaccination during the autumn–winter season, only 15 CVID patients were chosen for immunization. No significant differences were found between the two groups in terms of age and gender. The mean age of patients was 36.6 ± 2.03 years. All participants were included only after signing the informed consent form.

The investigation was permitted by the Research and Ethics Committee of the Institute of Immunology of the Federal Medical Biological Agency: approval number №11-1, 12 November 2018 and approval number №7, 8 July 2019.

### 2.3. Interventions

During the 2018-2019 flu season, the Department of Immunopathology of the Institute of Immunology of the Federal Medical and Biological Agency of Russia conducted a study involving 6 patients diagnosed with CVID who received a single dose (0.5 mL) of a quadrivalent subunit adjuvanted influenza virus vaccine (aQIV) “Grippol Quadrivalent”, manufactured by NPO Petrovax Pharm LLC, Russia. In the 2019–2020 flu season, 9 new CVID patients were immunized with a double dose (2 × 0.5 mL) of an inactivated subunit adjuvanted trivalent influenza virus vaccine (aTIV) “Grippol Plus” from the same manufacturer.

All participants had no previous IVV in the preceding two seasons (2016–2017, 2017–2018) or previous infections recorded in the past 6 months.

Inclusion criteria:Diagnosis of CVID confirmed in accordance with the diagnostic guidelines established by the European Society for Immunodeficiency Diseases and the American Academy of Allergy, Asthma, and Immunology for the diagnosis and management of IEI.IVIg therapy administered no more than 28 days before vaccination and no less than 21 days after, ensuring a minimum gap of 7 weeks between consecutive immunoglobulin doses.Secondary hypogammaglobulinemia causes ruled out.No use of glucocorticosteroids or other immunosuppressive therapies at the time of the study or within 3 months before initiation.No evidence of protein-losing enteropathy or suspicion of oncological or lymphoproliferative diseases in the CVID patients at the time of the study.Absence of specific antiviral antibodies at protective levels (above 1:40) in pre-vaccination blood samples.No vaccination against other infections within 1.5–2 months prior to study enrollment.All contraindications stated in the instructions for the vaccines.

The vaccines “Grippol Quadrivalent” and “Grippol Plus” were developed for people from 18 to 60 years of age for influenza prophylaxis. These vaccines included strains that were suggested by the World Health Organization (WHO) for the 2018–2019 and 2019–2020 seasons. “Grippol Quadrivalent” included strains such as A/Michigan/45/2015 (H1N1)pdm09-like virus, A/Singapore/INFIMH-16-0019/2016 (H3N2)-like virus, B/Colorado/06/2017-like virus (B/Victoria/2/87 lineage), and B/Phuket/3073/2013-like virus (B/Yamagata/16/88 lineage); and “Grippol Plus” in 2019–2020 included A/Brisbane/02/2018 (H1N1)pdm09-like virus, A/Kansas/14/2017 (H3N2)-like virus, and B/Colorado/06/2017-like virus (B/Victoria/2/87 lineage). Each vaccine contained 5 micrograms of hemagglutinin for every influenza virus strain along with 500 micrograms of azoximer bromide, preservative-free.

### 2.4. Outcomes

The analysis of blood samples was performed in the laboratory using certified equipment of the collective use center of the I. I. Mechnikov Research Institution of Vaccines and Serums in Moscow.

Blood samples were taken for analysis of immune response parameters at two time points: immediately on the day of vaccination before immunization and 24 ± 3 days after it. Expression of CD45+CD3−CD19+, CD45+CD3+CD19−, CD45+CD3+CD4+, CD45+CD3+CD8+, and CD45+CD3+CD16+CD56+ was measured by flow cytometry on the Cytomix flow cytometer FC500 (Beckman Coulter, Brea, USA) with the use of monoclonal antibodies (mAbs) Immunotech, Marseille, France. The leukocyte pool was gated based on CD45 markers, cell size, and granularity.

To avoid erroneous results, we used a calibrated positive displacement pipette to deliver biological samples and Flow-Count fluorospheres. Before use, we ensured that the fluorospheres were completely resuspended. To do this, Flow-Count fluorospheres were mixed on a vortex mixer for 10 to 12 s. A quantity of 100 µL Flow-Count Fluorospheres was added to the test tube, the contents of the test tube were mixed on a vortex mixer at 50% of the height of the test tube for 5 s. Vortex mixing was repeated immediately prior to flow cytometry analysis.

To obtain optimal results when adding 100 µL of the biological sample and 100 µL of the fluorospheres to the test tube, precise and careful pipetting techniques were necessary. To do this, we referred to the manufacturer’s pipetting instructions.

Thus, the biological sample was labeled with the desired antibody and lysed using the OptiLyse C reagent system. Prepared samples were analyzed on a flow cytometer within 2 h or less after the addition of the Flow-Count fluorospheres.

Then, the flow cytometer was properly aligned and standardized for light scatter and fluorescence intensity, and color compensation was set. To calibrate the flow cytometer, Flow-Check fluorosphere particles were used daily to check the optical system and inkjet automation of the cytometer. Histograms were created following the instructions on the package insert of the respective products and an additional two-parameter histogram of FL4 LOG (or FL3 LOG or FL2 LOG or FL1 LOG) to FS or TIME. A linear (or equivalent) selector was drawn around the Flow-Count fluorospheres (upper right corner) and designated CAL. We entered the analytical concentration of the Flow-Count fluorospheres as indicated on the Assay Data Sheet included with the package insert. To avoid erroneous results, we ensured that at least 1000 Flow-Count fluorospheres were counted. To determine absolute target cell counts on Beckman Coulter flow cytometers, the Flow-Count Fluorosphere Analytical Concentration under the CAL area designation was entered. Once at least 1000 fluorospheres were counted, the absolute number of target cells was automatically calculated using the following formula: absolute number (cells/µL) = total number of cells counted/total number of fluorospheres counted × Flow-Count Fluorosphere Analytical Concentration.

The expression level of TLRs on the peripheral immune cells (granulocytes, lymphocytes, monocytes) was analyzed both before vaccination and 24 ± 3 days after. This in vitro study utilized flow cytometry with monoclonal antibodies specific for TLR3-PE, TLR7-FITC, TLR8-FITC, and TLR9-PE (eBioscience, USA) on a Cytomix FC 500 flow cytometer (Beckman Coulter, USA), following the protocol provided in the manufacturer’s instructions.

### 2.5. Statistical Methods

For indicators of cellular immunity in the post-vaccination period, percentage and absolute content of lymphocyte subpopulations, medians, interquartile range, and 95% confidence interval for the median are given (Appendix A). Delta changes in percentage and absolute cell counts were calculated as the difference between the values after and before vaccination. Descriptive data were conveyed through the median along with its 95% confidence interval. The study analyzed trait dynamics and conducted group comparisons employing a robust linear mixed effects model (RLMM) [81]. The statistical significance of the model’s coefficients was computed using the Satterthwaite degrees of freedom approximation [82]. Post-hoc comparisons across groups at control points and within groups across control points were derived by generating suitable contrasts from the estimated model using the emmeans package [83]. The model analysis was based on transformed original data, utilizing arcsine transformation (for cell percentages) and logarithmic transformation (for absolute cell numbers). To compare the changes in cell counts between study groups, the Mann–Whitney test for independent samples (one and two doses of the vaccine) was applied. Differences were deemed statistically significant at *p* ≤ 0.05 and not significant at *p* ≥ 0.10; intermediate *p*-values (0.05 < *p* < 0.1) prompted discussions of potential trends. Calculations and visualizations were executed using both GraphPad Prism software (v.9.3.0, license GPS-1963924) and the R statistical environment (v.3.6, GNU GPL2 license).

## 3. Results

### 3.1. Changes in the Main Peripheral Blood Lymphocyte Subpopulation in Patients with CVID Vaccinated with One and Two (Simultaneous) Doses

A statistically significant difference in the percentage level of CD3+CD19− was revealed (t = 4.4, *p* < 0.001, df = 13). In the group of patients vaccinated with two doses simultaneously, a month after vaccination this indicator remained unchanged, while in the group of patients vaccinated with only one dose of the vaccine there was a statistically significant increase from 78.3 (73.4–83.5)% to 82.8 (75.6–87.7) % (*p* < 0.001) (Figure 1). The delta of changes (the difference between the parameters 24 ± 3 days after primary vaccination and before it) was +2.9 [0.9; 5.8]%, and in the group with two doses it was 0.8 [2.7; 0.1]%; the differences are statistically significant (*p* < 0.001) (Figure 2).

A similar situation is observed for the relative count of CD3−CD19+: statistically significant multidirectional dynamics were revealed in the study groups (t = 2.1, *p* = 0.05, df = 13). The level of CD3+CD19− increased as a result of vaccination with one dose of the vaccine while the level of CD3−CD19+, on the contrary, decreased (Figure 3). At 24 ± 3 days after vaccination in the group of patients vaccinated with one dose of the vaccine, there was a decrease in the percentage of CD3−CD19+ from 9.8 (6.7–14.3)% to 7.5 (5.1–11.0)% (*p* = 0.009). In the group of patients vaccinated with two doses, no such changes were detected. The delta change in the percentage of CD3−CD19+ was 1.9 [3.9; 0.3]% and with two doses +0.2 [0.9; 1.4]%; the differences are statistically significant (*p* = 0.03) (Figure 4).

### 3.2. Changes in the Expression of Toll-like Receptors in Patients with CVID Vaccinated with One and Two (Simultaneous) Doses

#### 3.2.1. Granulocytes

After one dose of the vaccine was administered, a statistically significant reduction was observed in the proportion of granulocytes expressing TLR9 (*p* = 0.05), and there was an almost statistically significant reduction in those expressing TLR3 (*p* = 0.08). Conversely, after two doses, the alteration in the proportion of granulocytes with TLR3 and TLR9 (measured before and 24 ± 3 days after vaccination) was 5.6% [9.8; 1.0] and 8.8% [12.2; 3.2] with one dose, and +3.5% [2.9; +19.4] and +4.5% [0.7; +26.5] following two doses, respectively (*p* = 0.11 for TLR3 and *p* = 0.03 for TLR9 when compared to a single dose) (Figure 5a,b). The percentage of granulocytes expressing TLR8 remained unchanged before and after vaccination, showing consistent dynamics regardless of whether one or two doses were administered.

#### 3.2.2. Lymphocytes

Following the administration of two doses of the vaccine, a notable rise in the percentage of lymphocytes expressing TLR9 was observed, which was statistically significant. This change was not seen after a single dose. The increase in the proportion of lymphocytes with TLR9 expression after two doses was +11% [+1.5; +29.4], compared to a minimal change of 0.1% [0.3; +0.3] after one dose, with the difference being statistically significant (*p* = 0.05) (Figure 6). On the other hand, the percentage of lymphocytes expressing TLR3 and TLR8 remained statistically unchanged regardless of whether one or two doses were administered, demonstrating analogous patterns with both dosing regimens.

#### 3.2.3. Monocytes

Administering two doses of the vaccine initially elevated the levels of monocytes that express TLR3 and TLR9. The rate of change in the proportion of monocytes featuring TLR3 and TLR9 was markedly distinct between the one-dose and two-dose groups, with both showing a *p*-value of 0.01 (Figure 7a,b). Although vaccination, whether with one or two doses, reduced the percentage of monocytes expressing TLR8, the extent of this reduction (delta percentage) did not significantly differ between the groups, with a *p*-value of 0.96.

## 4. Discussion

Researches have demonstrated that the absence of mature B cells does not influence the development of virus-specific memory CD4+ T cells, which might play a role in providing resistance to viral infections in mice even when B cells and CD8+ T cells are missing [84,85,86]. Examination of the functional activity of CD4+ T cells in response to viral antigens in individuals with agammaglobulinemia showed results comparable to those in healthy ones [87].

Currently, there is no universally accepted guideline or recommendation for evaluation T-cell immune responses to vaccination in humans. Typically, investigators measure cytokine production, antigen-induced proliferative responses, CD69, CD40L, or OX40, which are pro-activation markers [14,88,89]. It also remains unclear how different methods and assays adequately reflect actual T-cell response rates.

As for the indicators of cellular immunity, our results are consistent with the data available in the world research base. However, comparison of experimental data between studies presents serious difficulties due to the heterogeneity of the group of patients with CVID, different types of studied vaccines, as well as the indicators being determined, time frame for their determination, research methods, and so on.

If we compare in general the initially determined subpopulations of lymphocytes, then we can say that our results are consistent with the conclusions of other researchers from different countries. For example, we noted CD4+ lymphopenia, which was previously described in CVID patients as a decrease in the number of naïve CD4+ T-lymphocytes. Giovannetti et al. previously classified patients with CVID according to the number of naïve CD4+ T-lymphocytes. The detected CD4 lymphopenia dramatically altered the normal CD4/CD8 ratio in combination with an increase in CD8+ T cells, an increase of which is associated with the presence of long-term inflammation [90] caused by recurrent infections [74]. However, vaccination, according to the results of our work, did not statistically significantly lead to changes in lymphocyte parameters (CD45+CD3−CD19+, CD45+CD3+CD19−, CD45+CD3+CD4+, CD45+CD3+CD8+, CD45+CD3+CD16+CD56+); there was only an increase in relative counts (%) of CD45+CD3+CD19-lymphocytes and a decrease in relative counts (%) of CD45+CD3−CD19+ lymphocytes in the group of patients vaccinated with one dose, which is not confirmed by the data obtained in patients with CVID vaccinated with two doses. Most likely, these changes are associated with a small number of patients (six participants) and random fluctuations of indicators, taking into account the fact that the absolute values did not change significantly.

It is worth noting that in healthy patients immunized with a quadrivalent adjuvanted vaccine with a reduced amount of antigens down to 5 μg against all four strain-specific surface antigens, the vaccine proved its immunogenicity according to criteria such as seroprotection (≥70%) and seroconversion (≥40%) levels, and seroconversion factors (≥2.5), both 24 ± 3 days after vaccination and after 3 months.

Upon examining the dynamics of cellular immunity in individuals who received two vaccine doses, no noticeable alterations were observed in cellular immunity metrics or leukocyte counts. Nonetheless, our research evaluated the influenza vaccine’s efficacy in CVID patients by enhancing Toll-like receptor expression on immune cells. This was done by comparing a single dose of the quadrivalent adjuvanted influenza virus vaccine with two doses of the trivalent adjuvanted influenza virus vaccine over a 7-week period, without the inclusion of IVIG immunotherapy. The adjuvant in these vaccines was expected to initiate an immune response by stimulating innate immune mechanisms and offering a wider array of antigens compared to the monovaccine Pandemrix [49].

Toll-like receptors (TLRs) are instrumental in the immune system’s response to vaccines. These pattern recognition receptors (PRRs) are vital for eradicating pathogens [91]. They initiate signaling pathways necessary for early defense mechanisms through the binding of phagocytes or the activation of dendritic cells. These mechanisms also foster dendritic cell development, thereby governing secondary or adaptive immunity. TLR membrane proteins, conserved across various cell types, such as monocytes, phagocytes, dendritic cells, and certain B-cell subsets, activate B cells via different routes. TLRs oversee the production of pro-inflammatory cytokines, essential in responding to bacterial, fungal, and viral infections. For instance, TLR3 significantly contributes to the cross-priming of naive CD8 T cells, which then differentiate into cytotoxic T cells targeting virus-infected cells [92]. Activation of TLR2, TLR4, and TLR9 leads to diverse expressions of adhesion molecules [93,94,95].

Enhanced activity of TLR2, TLR3, TLR4, TLR7, and TLR9 stimulates dendritic cells, thereby boosting the activation of antigen-presenting cells through the induction of pro-inflammatory cytokines and the upregulation of costimulatory molecules necessary for antibody production [96]. Additionally, TLRs determine dendritic cell efficiency in antigen presentation, crucial for protecting immunocompromised individuals. Influenza virus vaccines activate innate immune cells, like myeloid and lymphoid dendritic cells, which serve as the initial defense against the virus [97].

Beyond bolstering innate immune responses, influenza virus vaccines enhance phagocytosis and possess antitoxic effects, such as reducing radical levels [98,99]. A 2022 review analyzed the safety and immunogenicity of a subunit influenza virus vaccine with the polymer adjuvant azoximer bromide, encompassing 11,736 participants aged from 6 months to 99 years, from 1993 to 2016 [100]. The use of influenza virus vaccines induced antibody production in children and adults under 60 during the transitional phase using traditional vaccines. Another study on azoximer bromide adjuvanted vaccines found that all evaluated vaccines (split, subunit, and adjuvanted) significantly increased the number of granulocytes expressing TLR2, TLR6, TLR8, and TLR9 in PBMC cultures compared to unstimulated cells. However, the adjuvanted vaccine had a higher induction potential for TLR9 and TLR8 than subunit and split vaccines, likely due to an adjuvant costimulatory effect.

We investigated the expression of TLR3, TLR8, and TLR9, which, like TLR7, are predominantly activated in endosomal compartments, enabling these receptors to detect viral and bacterial DNA and RNA degradation products. TLR9 is expressed on human B cells. Activation of TLR3 on PBMCs, including T, B, and NK cells, monocytes, dendritic cells, as well as fibroblasts in immunocompromised patients, results in normal IFN-α and IFN-β production, potentially ensuring adequate viral protection.

We gathered data on TLR activation in immune cells following single and double doses of vaccines. After the initial doses, a decrease in several parameters was noted (the proportion of granulocytes expressing TLR3 and TLR9, and monocytes expressing TLR3, TLR8, and TLR9). However, post-second doses showed an increase in certain parameters (the proportion of lymphocytes expressing TLR9 and monocytes expressing TLR3 and TLR9), indicating a significant protective effect against respiratory diseases, because stimulation of TLR 9 located on B-lymphocytes and TLR3 on lymphocytes (T, B and NK cells), dendritic cells and monocytes, as well as fibroblasts, is most likely responsible for the proper defense of organisms against pathogens (mainly viruses) due to their intracellular location and thanks to the normal production of IFN-α/β as a result.

## Figures and Tables

**Figure 1 vaccines-12-00843-f001:**
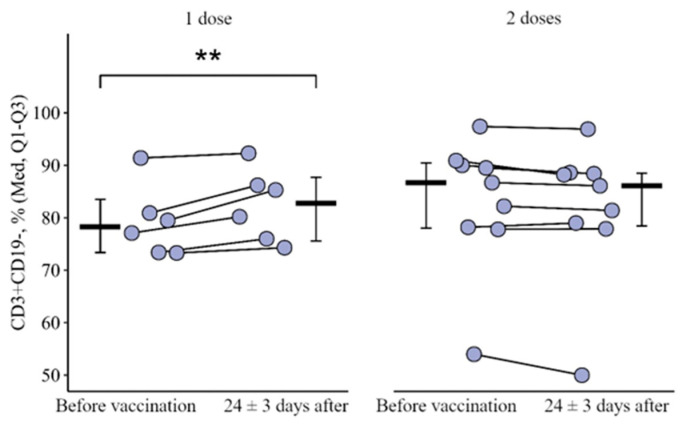
Content of CD3+CD19 lymphocyte subpopulations in the blood of patients with CVID depending on the stage of vaccination (before and after 24 ± 3 days) and administration of a single or double vaccine dose (individual values, median, and interquartile range are given). **—statistically significant differences before and after vaccination at *p* < 0.01; a robust linear mixed effects model was used for calculations.

**Figure 2 vaccines-12-00843-f002:**
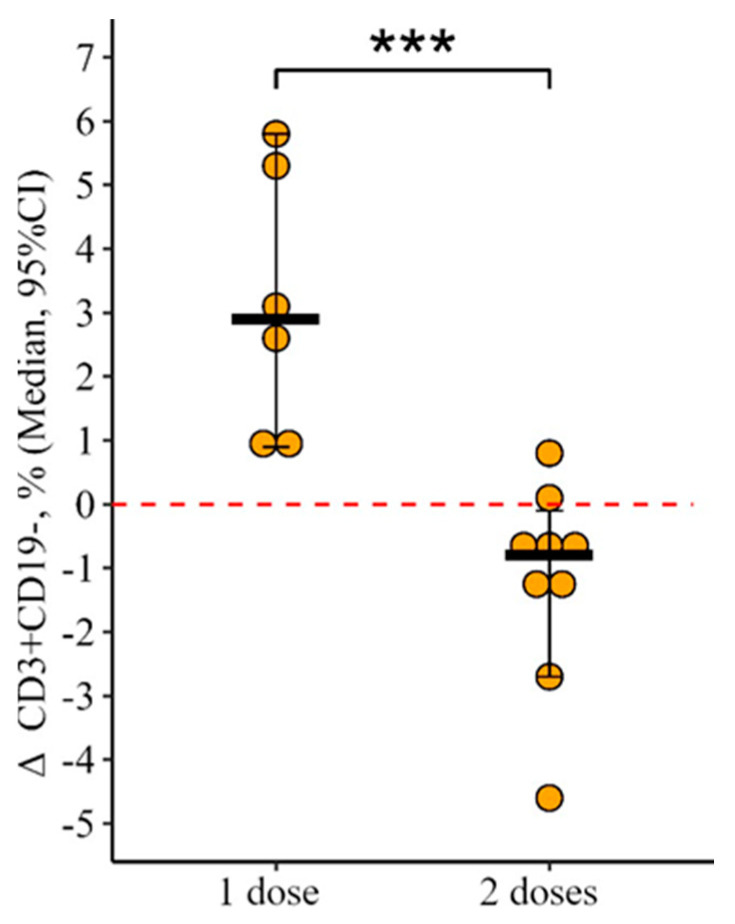
Delta of the content of subpopulations of CD3+CD19− lymphocytes in the blood of patients with CVID depending on the administration of a single or double vaccine dose (individual values, median, and its 95% confidence interval are given); ***—statistically significant differences between study groups at *p* < 0.001; the Mann–Whitney test was used.

**Figure 3 vaccines-12-00843-f003:**
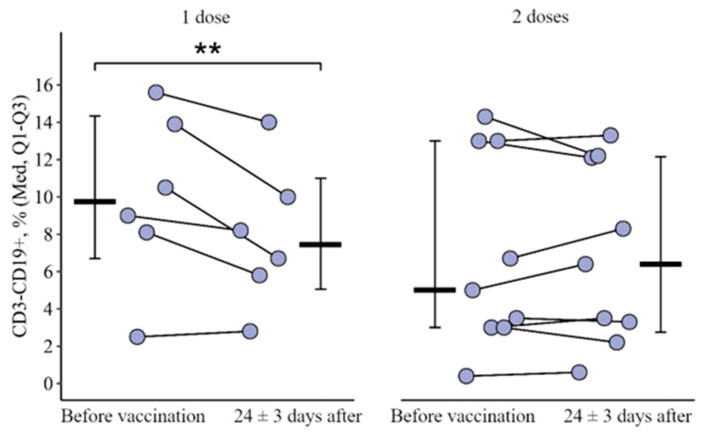
Content of CD3−CD19+ lymphocyte subpopulations in the blood of patients with CVID depending on the stage of vaccination (before and after 24 ± 3 days) and administration of a single or double vaccine dose (individual values, median, and interquartile range are given); **—statistically significant differences before and after vaccination at the *p* < 0.01 level; a robust linear mixed-effects model was used.

**Figure 4 vaccines-12-00843-f004:**
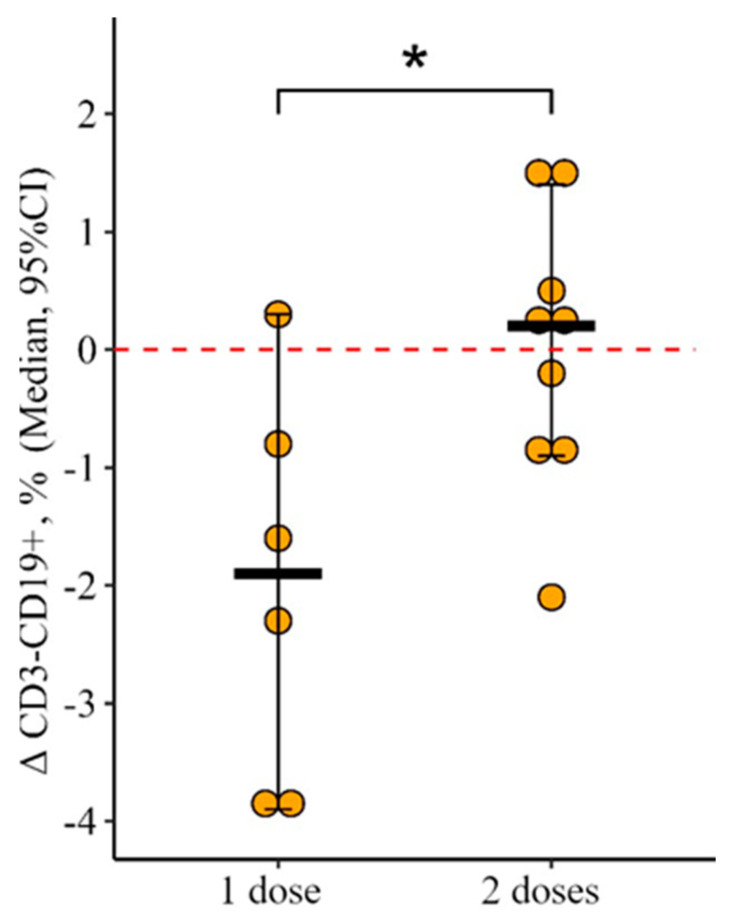
Delta of the content of subpopulations of CD3−CD19+ lymphocytes in the blood of patients with CVID depending on the administration of a single or double vaccine dose (individual values, median, and its 95% confidence interval are given); *—statistically significant differences between study groups at *p* ≤ 0.05 level; the Mann–Whitney test was used.

**Figure 5 vaccines-12-00843-f005:**
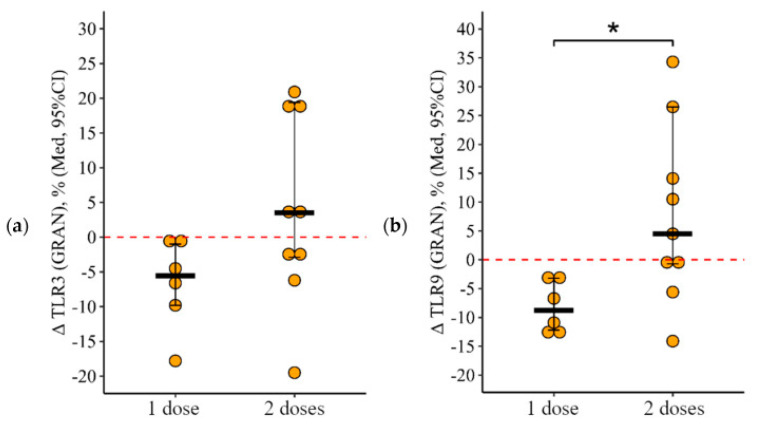
Change in the percentage of granulocytes showing TLR3 (**a**) and TLR9 (**b**) expression in CVID patients after vaccination with either single or double doses (individual data points alongside the median with a 95% confidence interval). *p*-value for the comparison of the change in granulocyte percentage expressing TLR3 and TLR9 following administration of one versus two doses of the vaccine. *—statistically significant differences between vaccination with one and two doses at the *p* ≤ 0.05 level. The Mann–Whitney test was used.

**Figure 6 vaccines-12-00843-f006:**
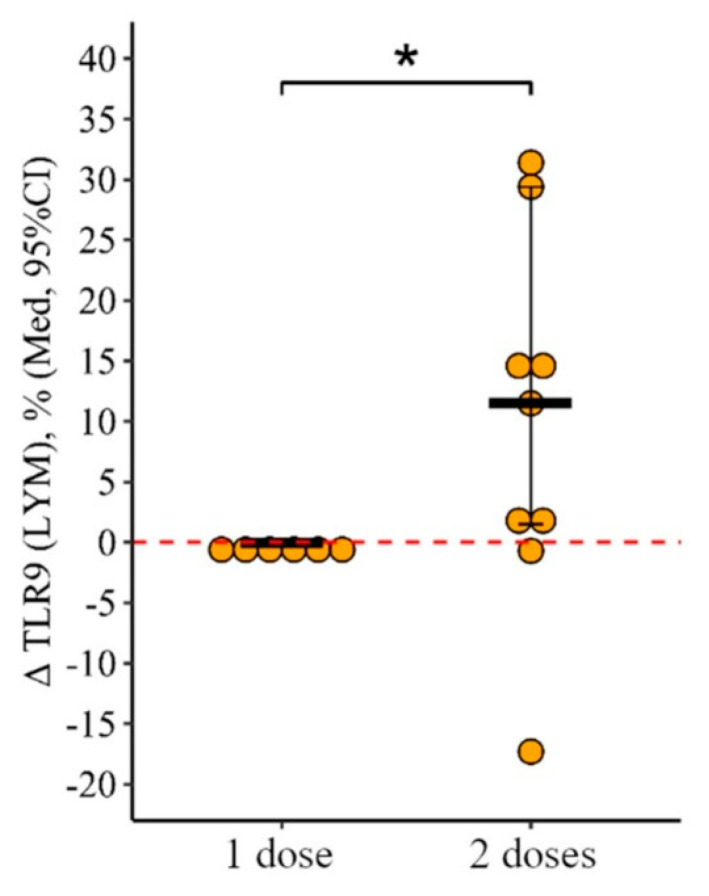
Change in the percentage of lymphocytes showing TLR9 expression in CVID patients after vaccination with either single or double doses (individual data points alongside the median with a 95% confidence interval). *—statistically significant differences between vaccination with one and two doses at the *p* ≤ 0.05 level. The Mann–Whitney test was used.

**Figure 7 vaccines-12-00843-f007:**
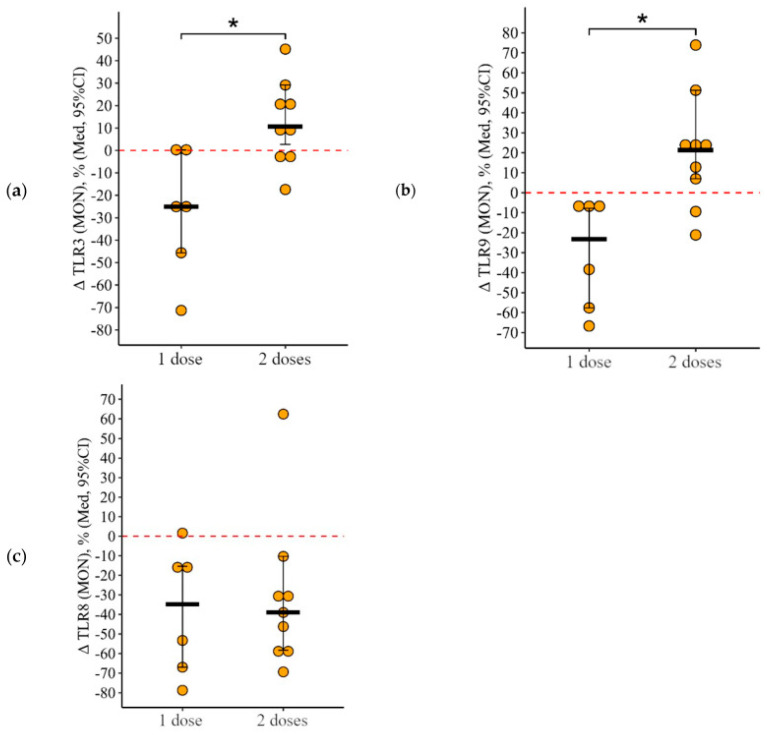
Change in the percentage of monocytes showing TLR3 (**a**), TLR9 (**b**), and TLR8 (**c**) expression in CVID patients after vaccination with either single or double doses (individual data points alongside the median with a 95% confidence interval). *p*-value for the comparison of the change in monocyte percentage expressing TLR3, TLR8, and TLR9 following administration of one versus two doses of the vaccine. *—statistically significant differences between vaccination with one and two doses at the *p* ≤ 0.05 level. The Mann–Whitney test was used.

## Data Availability

The authors declare that the data supporting the findings of this study are available within the paper and its Appendix A (p. 14).

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
