# Peer review of "Comparison of Post-Vaccination Cellular Immune Response in Patients with Common Variable Immune Deficiency"

_vaccines, 2024, doi:10.3390/vaccines12080843_

Round 1

Reviewer 1 Report

Comments and Suggestions for Authors

It is important to understand the cellular immunity of patients with inborn errors of immunity (IEI), especially those with antibody deficiencies. These authors evaluated the cellular immunity of six CVID patients who received one dose of a quadrivalent adjuvanted influenza vaccine, and nine patients were vaccinated with two doses of a trivalent inactivated influenza vaccine. They found an increase in the proportion of granulocytes (TLR3 and TLR9), lymphocytes (TLR3 and TLR8), and monocytes (TLR3 and TLR9), indicating the expression of toll-like receptors in these patients. This study marks the first instance of using simultaneous two-dose vaccination, which was associated with elevated levels of TLR expression in the immune cells. Administration of the adjuvanted vaccines in CVID patients appears promising. Their data supported the conclusions of the study. However, the following comments should be addressed before being accepted for publication.

1.      Lines 27-28: “The problem of the vaccine-specific T-cell response identifying is still a matter of debate:” The meaning of this sentence needs to be clarified.

2.      Line: 317-320: Expression of СD45+СD3-СD19+, СD45+СD3+СD19-, СD45+СD3+СD4+, СD45+СD3+СD8+, СD45+СD3+СD16СD56+ was measured by flow cytometry on the Cytomix flow cytometer FC500 (Beckman Coulter, USA) with the use of monoclonal antibodies (mAbs). СD45+СD3+СD16СD56+ or СD45+СD3+СD16+СD56+? Why are these markers chosen, and how are they used to calculate the data shown in Figures 1 and 2?

3.      Lines 119-120: What monoclonal antibodies were used, and the source of each monoclonal antibody should be described.

4.      Lines 323-324: How was the flow cytometer calibrated for quantitative analysis should be described.

5.      Lines 348-349: A statistically significant difference in the percentage level of CD3+CD19- was revealed. How this is calculated should be described.

6.      Lines 464-471: “However, vaccination, according to the results of our work, did not statistically significantly lead to changes in lymphocyte parameters (CD3+, CD4+, CD8+, 465 CD16,56+, CD19+):” Is this consistent with the results shown in Figures 1-4? Please clarify.

7.      Lines 425-430:  ” Administering two doses of the vaccine initially elevated the levels of monocytes that 425 express TLR3 and TLR9. The rate of change in the proportion of monocytes featuring TLR3 426 and TLR9 was markedly distinct between the one-dose and two-dose groups, with both 427 showing a p-value of 0.01 (fig. 7a, b). Although vaccination, whether with one or two doses, 428 reduced the percentage of monocytes expressing TLR8, the extent of this reduction (delta 429 percentage) did not significantly differ between the groups, with a p-value of 0.96.” This is a very interesting observation. It would be much more interesting if a possible explanation of these observations were given or a hypothesis of the possible cause of the differences would significantly increase the interesting level of this manuscript.

Reviewer 2 Report

Comments and Suggestions for Authors

According to the authors, this study aimed to assess the effect of inactivated adjuvanted subunit influenza vaccination on the expression of TLRs on the immune cells of patients with common variable immunodeficiency.  Although, in a general way, the authors presented interesting data, some points should be considered.

In Abstract section

1) Please add “TLRs” after “toll-like receptors” in line 31.

2) Please alter the use of "expression levels" to mention "key lymphocyte subpopulations". It is correct to use "proportion" or "number".  Suggestion: "The proportion of key lymphocyte subpopulations and expression levels of TLRs were analyzed using flow cytometry with monoclonal antibodies."

3) Based on the results obtained in this study, it is necessary to alter the description of the main findings in this section to "However, after vaccination, higher expression of TLR3 and TLR9 in granulocytes, monocytes, and lymphocytes was found in those patients who received two vaccine doses than one single dose."

In Introduction section

4) Please add the word “virus” after Influenza, especially related to the Influenza vaccination. To be clear, use the term “Influenza virus vaccination” and the abbreviation IVV throughout the text.

5) Page 4, line 203, what is the abbreviation "PAMPs"?

6) Since one of the main findings is associated with TLR3, it is appropriate to present the importance of this type of receptor for immunity.

7) Despite the authors presenting the purpose of the study in the next section (Material and Methods), they not only should place this description but also provide a better presentation of the relevance of the present study, as the last sentence of this section.

Material and Methods section

8) In order to better present the data in this section, the authors should structure this section based on The Transparent Reporting of Evaluations with Nonrandomized Designs (TREND) [doi: 10.2105/AJPH.94.3.361].

9) A representative study design scheme could be useful to improve the meaning of the study.

10) What was the age of the participants in this study?

11) Please provide the approval number of the Research and Ethics Committee.

12) Please add the symbol "-" or "+" after CD16, in line 318 (СD45+СD3+СD16СD56+). In addition, please clarify how these different cell labeling strategies were used to define each cell population assessed here.

13) In line 321, the authors mentioned that they assessed "The presence of TLRs(...)", but the correct is to mention "The expression level of TLRs(...)".

Results section

14) The presentation of the effect size values, in some evaluations, can improve the significance of the findings described here.

15) No data concerning the values of the percentage or even the absolute number of leukocytes assessed here (granulocytes, monocytes, and especially CD4+ and CD8+ T lymphocytes), were provided. by the authors. Since these data are relevant to this study, they need to present them as Figures or a Table, even as supplementary material.

16) Although it was not aimed at the study, data regarding the antibody response to the IVV in the volunteer groups is very important to be shown and can improve the meaning of the present study. I believe that the authors have these results and can add to this manuscript.

Discussion section

17) I suggest initiating this section summarizing the main results obtained in the study.

18) The authors mentioned and focused one significant part of the discussion on the relevance of T cells (both CD4+ and CD8+) in the study context, but no data regarding these specific cells was shown here.

19) In addition, they mentioned in lines 468-469 "(...) as well as by maintaining similar values in these patients a year later when the dose was repeated." What does it mean? Where were these data shown?

20) Please correct this information - (CD3+, CD4+, CD8+, CD16,56+, CD19+). Lines 465-466.

Round 2

Reviewer 2 Report

Comments and Suggestions for Authors

As the authors adequately answered all questions and carefully corrected the manuscript, I recommend that the MS be accepted for publication.